# Identifying interactions across brain areas while accounting for individual-neuron dynamics with a Transformer-based variational autoencoder

**Qi Xin**
Carnegie Mellon University
Pittsburgh, PA 15213
xinqi0511@gmail.com

**Robert E. Kass**
Carnegie Mellon University
Pittsburgh, PA 15213
kass@stat.cmu.edu

## Abstract

Advances in large-scale recording technologies now enable simultaneous measurements from multiple brain areas, offering new opportunities to study signal transmission across interacting components of neural circuits. However, neural responses exhibit substantial trial-to-trial variability, often driven by unobserved factors such as subtle changes in animal behavior or internal states. To prevent evolving background dynamics from contaminating identification of functional coupling, we developed a hybrid neural spike train model, GLM-Transformer, that incorporates flexible, deep latent variable models into a point process generalized linear model (GLM) having an interpretable component for cross-population interactions. A Transformer-based variational autoencoder captures nonstationary individual-neuron dynamics that vary across trials, while standard nonparametric regression GLM coupling terms provide estimates of directed interactions between neural populations. We incorporate a low-rank structure on population-to-population coupling effects to improve scalability. Across synthetic datasets and mechanistic simulations, GLM-Transformer recovers known coupling structure and remains robust to shared background fluctuations. When applied to the Allen Institute Visual Coding dataset, it identifies feedforward pathways consistent with established visual hierarchy. This work offers a step toward improved identification of neural population interactions, and contributes to ongoing efforts aimed at achieving interpretable results while harvesting the benefits of deep learning.

## 1 Introduction

A central goal of systems neuroscience is to understand how multiple populations of neurons, across different brain areas, interact as a coordinated circuit to produce perception and behavior. Recent advances in electrophysiological recording technologies, which have enabled simultaneous recordings from thousands of neurons across anatomically distinct parts of the brain, have offered exciting new opportunities for progress [1, 2, 3, 4]. Because neural action potentials, or spikes, are the dominant sources of information transmission, much of modern neurophysiology has focused on sequences of spikes, that is, neural spike trains. Neural networks in living animals, however, also exhibit continually evolving dynamics. Thus, a major contemporary challenge is to develop methods for identifying interactions among spike trains from large populations of neurons while accounting for changing background dynamics.

One of the main collections of methodologies for analyzing neural spike trains has come under the heading of generalized linear models (GLMs). These models are based on Poisson nonparametric regression and often incorporate latent variables. They have been remarkably adept at identifying functional interactions between neurons due to their flexibility, interpretability, and rigorous statistical

foundations [5, 6, 7, 8, 9, 10]. In these models, a neuron's spiking activity is treated as a point process whose intensity function depends on behaviorally relevant variables such as external stimuli, the recent spiking history of the neuron itself, and also the recent spiking history of other neurons. The latter is used to represent interaction effects as coupling terms in the intensity function.

GLMs typically use a trial-invariant function of time, such as a function of time since stimulus onset, to model nonstationary effects that are not explicitly attributed to cross-neuron coupling or the neuron's own spike history [5, 6, 7, 8, 11]. However, even under identical stimulus conditions, individual-neuron dynamics often exhibit substantial trial-to-trial variability [12, 13, 14, 15, 16]. This structured variability can be driven by behaviors such as locomotion, which induce strong and widespread background fluctuations in firing rates throughout the brain, including in primary sensory areas [17, 18, 19, 20, 21, 22, 23, 24, 25, 26, 27, 28]. Endogenous states such as arousal or attention can also contribute to trial-to-trial variability and are often difficult to measure precisely [12, 25, 28]. Additional sources of within-area variability, possibly arising from unobserved factors, may further complicate the individual-neuron dynamics. Although scaling the firing rates of a trial with a trial-wise gain constant, or including observed behavioral covariates in the model, may help mitigate effects due to trial-to-trial variability, they may be too simple or require careful hands-on model engineering.

Deep artificial neural networks, which can capture complicated nonlinear relationships [29, 30], have become popular for modeling large-scale, multi-area neural activity [31, 32, 33, 34, 35, 36, 37, 38, 39, 40]. They are particularly attractive for predictive purposes, but lack the straightforward interpretability of GLMs. In this article, we introduce a hybrid framework we call GLM-Transformer. It combines GLM-based coupling to identify cross-area interactions, with a Transformer-based variational autoencoder (VAE) to capture individual-neuron dynamics that do not require interpretation (so these dynamics act as high-dimensional nuisance parameters, in the terminology of statistics). Following previous neural applications of deep learning, we incorporate dimensionality reduction [31, 32, 40] in the spirit of Gaussian Process Factor Analysis (GPFA) [41], where a small number of latent factors drive the activity of tens or hundreds of neurons. We demonstrate that our method can accurately recover coupling effects in synthetic datasets, generated from both statistical GLM neurons and mechanistic exponential integrate-and-fire (EIF) models, while alternative methods do not. We then apply our GLM-Transformer to the Allen Institute's Visual Coding dataset [42], where it identifies strong directional coupling from the primary visual area V1 to the lateral medial (LM) and anterolateral (AL) visual areas, in agreement with prior studies. Together, these results illustrate the utility of combining deep artificial neural networks with interpretable statistical models in identifying cross-area interactions from large-scale neural recordings.

## 2   Related work

Some recent advances, including LFADS [31], NDT [33], NDT2 [34], MtM [36] and Meta-Dynamical State Space Models [43], aim to learn meaningful latent representations at each time bin that generate observed activity. In contrast, Deep Random Splines (DRS) [37] aims to learn trial-level latent representations, which capture trial-to-trial variability. While these models offer powerful representations and predictions, they do not easily identify interactions across neural populations.

Several approaches based on state-space models have accommodated neural interactions, including mp-srSLDS [44], MR-SDS [40], while STNDT [39] treats neurons as tokens in a Transformer architecture. In these models, interactions occur in latent space, and in MR-SDS and STNDT, interactions are parameterized by deep networks. They again do not provide direct interpretation of the way activity in one population affects activity in another.

## 3   GLM-Transformer

**Overview**   The GLM-Transformer decomposes drivers of each neural population's activity into three components: individual-neuron dynamics, the population's self-history effects, and cross-population coupling effects. Individual-neuron dynamics replace the baseline term or stimulus effect in the traditional GLM framework in order to capture not only standard experimental effects, but also trial-to-trial variability in background dynamics arising from other sources, including subtle changes in behavior and endogenous states. Because background fluctuations are often correlated across

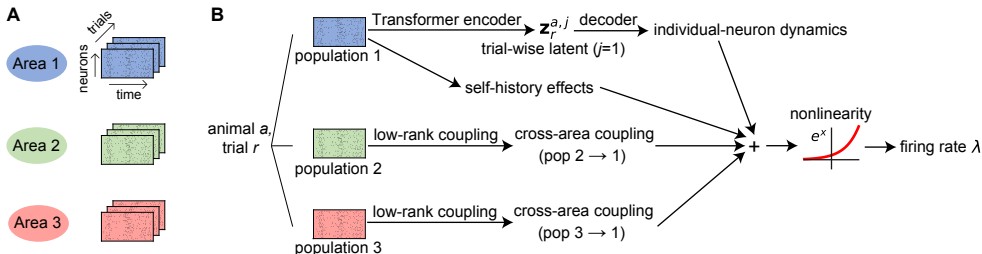

Figure 1: Illustration of data and GLM-Transformer model architecture. (**A**) Neural activity is recorded across multiple brain areas, with data from each area organized as a three-dimensional array of neurons, time, and trials. (**B**) Schematic of GLM-Transformer. For each trial $r$ and animal $a$, a Transformer encoder processes spike trains from a target population to produce a trial-wise latent variable $\mathbf{z}_r^{a,j}$, which is decoded into smooth individual-neuron dynamics. The predicted firing rate is computed by combining individual-neuron dynamics with self-history effects from the population itself and coupling terms from other populations, followed by a nonlinearity.

areas [15, 19, 20, 27, 28], explicitly modeling these shared dynamics helps prevent misattribution of such variability to cross-area coupling and improves robustness of coupling identification. The self-history component accounts for recurrent dynamics within each population and each neuron's own post-spike effect (e.g., its refractory period). The cross-population coupling component aims to capture dependencies between neural populations while accounting for the other components in the model.

GLM-Transformer supports training across multiple animals or recording sessions. In animal $a$ and neural population or brain area $j$, let $Y_{r,n,t}^{a,j} \in \mathbb{N}$ denote the spike count of neuron $n$ at time bin $t$ in trial $r$. Here, $j = 1, 2, \ldots, P$, where $P$ is the total number of populations. We assume spike counts follow a Poisson distribution, with the mean corresponding to the underlying firing rate at that time point: $Y_{r,n,t}^{a,j} \sim \mathrm{Poisson}(\lambda_{r,n,t}^{a,j})$. Throughout the paper, we use a bin size of 2 ms and rarely observe more than one spike per bin. To emphasize the smooth nature of firing rates over time, we use the notations $\lambda_{r,n}^{a,j}(t)$ and $\lambda_{r,n,t}^{a,j}$ interchangeably.

Inspired by Deep Random Splines [37], we model individual-neuron dynamics with a trial-wise $d_z$-dimensional latent variable $\mathbf{z}_r^{a,j} \in \mathbb{R}^{d_z}$ that captures trial-to-trial variability. The log-intensity of neuron $n$ in population $j$ is modeled as

$$\log \lambda_{r,n}^{a,j}(t) = f_n^{a,j}(\mathbf{z}_r^{a,j}, t) + \sum_{i=1}^{P} c_{r,n}^{a,i \to j}(t) + h_{r,n}^{a,j}(t), \tag{1}$$

where the individual-neuron intensity function $f_n^{a,j}(\mathbf{z}_r^{a,j}, t)$ depends on both time $t$ and the trial-wise latent variable $\mathbf{z}_r^{a,j}$; it is modeled as a low-dimensional smooth function constructed from B-spline basis functions whose coefficients are produced by a Transformer-based encoder-decoder.

The coupling term $c_{r,n}^{a,i \to j}(t)$ captures the effect of spikes from population $i$ on neuron $n$ in population $j$ through a low-rank structure, implemented as a linear combination of convolutions between spike trains from population $i$ and learned temporal filters parameterized by raised cosine basis functions [6]. Raised cosine basis functions concentrate density at short time lags and provide an inductive bias that better captures rapid post-spike and coupling effects compared with the smoother and more uniform B-spline basis.

The final term, $h_{r,n}^{a,j}(t)$, represents the neuron's own post-spike effect and is computed by convolving the single-neuron spike train with a post-spike filter. Together, $h_{r,n}^{a,j}(t)$ and the within-population component of the coupling term (when $i = j$) constitute the self-history effect for population $j$. To avoid non-identifiability, we penalize roughness in the background dynamics so that they vary more slowly than the coupling effects. In the following sections, we describe each component of the model in detail.

Although GLM-Transformer models multiple sources of variability, its primary goal is to identify directed interactions between neural populations. The trial-specific latent variables and individual-

neuron dynamics control for background fluctuations that could obscure these interactions, enabling more accurate and interpretable estimates of cross-area coupling.

**Encoder**  The goal of the encoder network is to estimate the trial-wise latent variable $\mathbf{z}_r^{a,j}$ for population $j$ based on the neural activity observed in that trial. Since $\mathbf{z}_r^{a,j}$ is unobserved, we use a variational inference framework. We place a standard Gaussian prior over the latent, $\mathbf{z}_r^{a,j} \sim \mathcal{N}(0, I)$, and approximate its posterior with a Gaussian distribution whose mean and variance are predicted by a Transformer-based encoder, conditioned on the spike trains from the trial:

$$q\left(\mathbf{z}_r^{a,j} \mid \mathbf{Y}_{r,:,:}^{a,j}\right) = \mathcal{N}\left(\mu_\phi(\mathbf{Y}_{r,:,:}^{a,j}), \mathrm{diag}(\sigma_\phi^2(\mathbf{Y}_{r,:,:}^{a,j}))\right), \tag{2}$$

where $\mathbf{Y}_{r,:,:}^{a,j}$ is the full spike train matrix of population $j$ in trial $r$, with neurons and time spanning the two axes; $\mu_\phi$ and $\sigma_\phi^2$ are the outputs of the Transformer-based encoder $\phi$, representing the mean and variance of the approximate posterior.

We use 10 or 20 ms time bins as tokens for the Transformer (which are wider than the 2 ms bins used elsewhere in the model). These bin widths were selected through hyperparameter tuning, which showed that performance was stable within this range and degraded slightly for smaller or larger bins. Because the set of recorded neurons differs across animals, we apply an animal-specific linear transformation to map the spike count vector in each time bin to a shared $d_{\mathrm{model}}$-dimensional embedding space. To provide temporal context, we include elementwise summation with positional encodings. The resulting sequence is passed through a stack of Transformer layers. Consistent with prior work, we found no benefit in using more than two Transformer layers or attention heads [33]. Finally, we obtain a summary embedding via average pooling over all tokens, which is then passed through a linear layer to produce the mean vector and variance matrix (which is diagonal) of the approximate posterior on the trial-wise latent vector for each population.

**Decoder**  The decoder uses the trial-wise latent $\mathbf{z}_r^{a,j}$ to produce a representation of individual-neuron dynamics. As in previous generalized factor analysis models [31, 32, 40, 41], we assume that the individual-neuron dynamics of a population are governed by a set of underlying low-dimensional individual-neuron dynamics factors, $\tilde{f}_l^{\mathrm{ind},a,j}(\mathbf{z}_r^{a,j}, t)$, each varying continuously over time for a given trial. Here, "ind" indicates individual-neuron dynamics, and the index $l = 1, 2, \ldots, L^{\mathrm{ind}}$ enumerates the $L^{\mathrm{ind}}$ factors. These factors are B-spline functions, where the trial-wise latent variable $\mathbf{z}_r^{a,j}$ is mapped to the B-spline coefficients via a feedforward network (a multilayer perceptron, MLP) with one hidden layer. To encourage smoothness, we apply an $\ell_2$ penalty on the second derivative of each factor function. In practice, we typically use 10 or 20 B-spline basis functions. The $l$-th individual-neuron dynamics factor function is computed as

$$\tilde{f}_l^{\mathrm{ind},a,j}(t) = F_{\mathrm{B\text{-}splines}}\left(\left[F_{\mathrm{MLP}}(\mathbf{z}_r^{a,j})\right]_l, t\right), \tag{3}$$

where $F_{\mathrm{MLP}}(\mathbf{z}_r^{a,j}) \in \mathbb{R}^{L \times K}$ maps the trial-wise latent to a matrix of B-spline coefficients for all factors, and $[\cdot]_l$ denotes the extraction of the coefficients corresponding to the $l$-th factor. $F_{\mathrm{B\text{-}splines}}$ then applies these specific coefficients to the basis functions. To get the individual-neuron dynamics intensity function for each neuron, we apply a linear readout from these factors:

$$f_n^{a,j}(\mathbf{z}_r^{a,j}, t) = \sum_{l=1}^{L^{\mathrm{ind}}} W_{l,n}^{\mathrm{ind},a,j} \tilde{f}_l^{\mathrm{ind},a,j}(\mathbf{z}_r^{a,j}, t), \tag{4}$$

where $W_{l,n}^{\mathrm{ind},a,j}$ are learnable readout weights that linearly combine the factors to produce the final individual-neuron dynamics intensity for neuron $n$ in population $j$.

**Low-rank population-to-population coupling effects**  To reduce the number of coupling parameters, which could be quadratic in the number of neurons, we impose a low-rank structure on the coupling effects between neural populations. Specifically, the influence of source population $i$ on target population $j$ is mediated through a set of low-dimensional coupling factors:

$$\tilde{f}_l^{\mathrm{coupling},a,i \to j}(t) = \sum_{n=1}^{N^{a,i}} \alpha_{l,n}^{a,i \to j}\left(g_l^{a,i \to j} * Y_{r,n}^{a,i}\right)(t), \tag{5}$$

where $N^{a,i}$ is the number of neurons in source population $i$ for animal $a$, and $l = 1, 2, \ldots, L^{\text{coupling}}$ indexes the $L^{\text{coupling}}$ coupling factors. The temporal filter $g_l^{a,i \to j}$ is learnable and parameterized using smooth raised cosine basis functions [6], and the convolution $*$ is applied to the spike train $Y_{r,n}^{a,i}$ of neuron $n$. The sending weight $\alpha_{l,n}^{a,i \to j}$ modulates the contribution of neuron $n$ to the $l$-th coupling factor. We have found it sufficient, in practice, to use only one factor for each coupling (i.e., $L^{\text{coupling}} = 1$). The coupling effect on the target population is then

$$\mathbf{c}_{r,n}^{a,i \to j}(t) = \sum_{l=1}^{L^{\text{coupling}}} W_{l,n}^{\text{coupling},a,i \to j} \, \tilde{f}_l^{\text{coupling},a,i \to j}(t), \tag{6}$$

where $\mathbf{W}_{l,n}^{\text{coupling},a,i \to j}$ is the receiving weight that maps coupling factor $l$ to neuron $n$ in the target population $j$. We add a small $\ell_1$ penalty $(1 \times 10^{-5})$ on both the sending and receiving weights during training.

**Self-history effects**   Self-history effects are modeled by combining within-population coupling and neuron-specific post-spike filters. The within-population coupling from population $i$ to itself is handled using the same low-rank structure described for cross-population interactions $(i \to i)$. Additionally, each neuron has its own post-spike filter that captures history-dependent modulation, such as refractory periods. These post-spike filters are also parameterized using a set of smooth raised cosine basis in a time delay window of 10 ms [6, 45], and are convolved with the neuron's own spike train.

**Training**   Each dataset is split into training (70%), test (10%), and validation (20%) sets. We used a four-stage training procedure that progressively incorporates model components: (1) train only the trial-invariant individual-neuron dynamics, without other components, using maximum likelihood; (2) enable the VAE and train the trial-varying individual-neuron dynamics with evidence lower bound (ELBO); (3) include population coupling effects and continue training with ELBO; (4) include neuron-specific post-spike filters and fine-tune the full model. This staged approach improves convergence and avoids poor local optima (compared with training all components together). We used Bayesian optimization for hyperparameter tuning. Full details about training and hyperparameter tuning are provided in Appendix E. All experiments were run on a single NVIDIA GeForce RTX 4090 GPU (24 GB) using PyTorch 2.5.1 with CUDA 12.1. Training on the Allen Institute dataset from ten animals took approximately 15–20 hours.

**Identifiability**   If the individual-neuron dynamics $f_n^{a,j}(\mathbf{z}_r^{a,j}, t)$ are too flexible, the model can become non-identifiable: the dynamic components may absorb variation that should be attributed to coupling effects. To avoid this, we made a strong assumption that coupling effects vary more rapidly than the background fluctuations arising from changes in behavior or endogenous states. This assumption is motivated by empirical observations that slower fluctuations in neural activity tend to have higher spectral power [12] and are often driven by behavioral and arousal-related factors evolving over hundreds of milliseconds, whereas cross-area interactions typically occur at much shorter latencies [46]. This is why we regularized the individual-neuron dynamics and used a low-dimensional trial-wise latent variable $\mathbf{z}_r^{a,j}$. Also, because cross-area coupling often involves only a small subset of neurons within a population [47, 46, 48], the loading patterns of individual-neuron dynamics factors and coupling factors typically differ, which further helps to disentangle these two components. When GLM-Transformer is reconfigured using a large number of B-spline basis functions, no smoothness penalty, a high-dimensional trial-wise latent variable $\mathbf{z}_r^{a,j}$, and an increased number of factors for the individual-neuron dynamics term, the troublesome absorption of coupling effects by individual-neuron dynamics can occur. See Supplementary Figure S1.

**Summary of model components**   To clarify the methodological contribution, we summarize how GLM-Transformer integrates and extends prior approaches. The individual-neuron dynamics component builds on ideas from Deep Random Splines (DRS) [37] and Neural Data Transformer (NDT) [33]. It combines the latent-per-trial structure and spline-based decoder from DRS with the flexible Transformer encoder from NDT, allowing trial-specific background dynamics to be modeled with both flexibility and structure. The low-rank GLM-style coupling component represents a new integration: although inspired by reduced-rank regression, it extends this idea to point process GLMs for estimating directed cross-area interactions in an interpretable way. The low-rank structure is

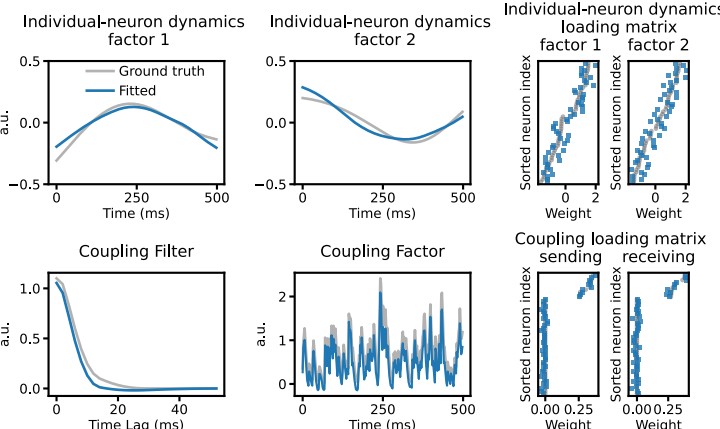

Figure 2: Recovery of ground truth components in synthetic data generated from GLM spiking neurons. Gray: ground truth; blue: fitted. Top row: ground truth versus fitted individual-neuron dynamics factors of an example trial (left, middle) and their loading matrix (right). Bottom row: ground truth versus fitted coupling filter (left), coupling factor of an example trial (middle), and sending/receiving weights of the coupling term (right).

crucial as it reduces model complexity and enable GLM-Transformer to scale efficiently to large neural populations while preserving the interpretability of coupling estimates. The post-spike history term follows established GLM formulations. Taken together, these elements form a unified framework that combines the interpretability of GLMs with the representational power of deep latent models, providing a principled and scalable approach for disentangling background dynamics from true cross-area coupling.

# 4 Experiments and results

## 4.1 Ablation study

We provide an ablation study (Table S1) in the Appendix A that evaluates the contributions of key components. These results confirm that each design choice improves model performance on both synthetic and real data.

## 4.2 Validation on synthetic data from GLM neuron models

We first evaluated whether GLM-Transformer can recover the ground truth structure when data are generated from a known GLM model. We simulated spiking activity from two neural populations, each containing 50 neurons. The individual-neuron dynamics (i.e., baseline firing rates) were generated using two latent factors per population, drawn from Gaussian processes with a temporal correlation structure defined by a 300-ms time constant. These factors were linearly combined using different loading weights for each neuron, mimicking structured trial-to-trial variability from unobserved sources. Cross-population coupling was introduced from the first population to the second using the same low-rank structure assumed by GLM-Transformer. Nonzero sending and receiving weights were assigned to only the first 10 neurons in each population, reflecting experimental findings that only a subset of neurons participate in cross-area interactions [47, 46, 48]. Spike trains were generated via an inhomogeneous Poisson process, with conditional log-intensity given by the sum of the individual-neuron dynamics, cross-population coupling, and a neuron-specific post-spike history term. The data set contained 2,000 trials, each 500 ms long, with an average firing rate of 26 spikes/sec.

Fitting GLM-Transformer to this synthetic dataset, it successfully recovered the ground truth. As shown in Figure 2, the estimated individual-neuron dynamics and, more importantly, the cross-population coupling match the true underlying components. The fitted post-spike filters also align

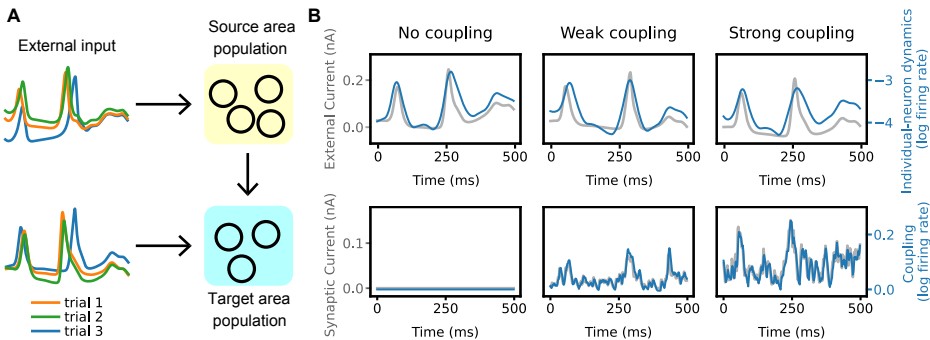

Figure 3: Comparison of ground truth synaptic currents with fitted components from GLM-Transformer in EIF neuron simulations. (**A**) Schematic of the simulation setup: external input drives both source and target populations, with synaptic connections from source to target neurons. (**B**) Each column shows results from one example trial under a different coupling condition: no coupling (left), weak coupling (middle), and strong coupling (right). Top row: external input (gray, left axis) and fitted individual-neuron dynamics (blue, right axis), averaged across all neurons in the target population. Bottom row: ground truth synaptic input from source to target (gray, left axis) and fitted coupling effect (blue, right axis), averaged over the 10 target neurons receiving cross-population input. Fitted coupling effects closely match the true synaptic currents across conditions.

with the ground truth (Supplementary Figure S2). These results validate both the accuracy of the GLM-Transformer and the training procedure.

### 4.3   Validation on simulated data from mechanistic neuron models

To test the robustness of GLM-Transformer on more realistic data, we simulated spike trains from two interconnected neural populations using an exponential integrate-and-fire (EIF) model [49, 50, 51] (see Appendix F for details). External inputs had a dual-peaked temporal shape, mimicking responses to drifting gratings in mouse visual cortex from the Allen Institute Visual Coding dataset, with peaks around 50 ms and 250 ms after stimulus onset [46]. Trial-varying gain modulations, sampled from a Gaussian process, were added to mimic behavioral fluctuations. Each population contained 50 neurons, with excitatory synapses from 10 neurons in the first population to 10 in the second, and log-normally distributed synaptic strengths [52]. Recurrent connections were also included within each population. We varied the mean cross-population synaptic strength to assess whether GLM-Transformer could recover the underlying coupling.

Figure 3 shows that GLM-Transformer produces interpretable components that qualitatively match the underlying biophysical ground truth across varying coupling strengths. Although the fitted individual-neuron dynamics do not exactly match the external input due to enforced smoothness, they capture the main temporal features, and more importantly the fitted coupling effect matches the ground truth synaptic input in all three conditions. To further evaluate scalability, we simulated a larger network with four interacting EIF populations and we found GLM-Transformer is still able to recover the true directed connections (Supplementary Figure S3). These results demonstrate the reliability of the GLM-Transformer while highlighting an advantage of GLM-based approaches, which is that the fitted components can reflect underlying currents or voltage signals [50, Section 10.2.1].

### 4.4   Robustness to shared background dynamics

A key motivation for GLM-Transformer is to avoid spurious coupling estimates in the presence of background gain fluctuations that are shared across neural populations. To evaluate this, we simulated two EIF neuron populations as in the previous experiment, but introduced correlated nonstationary gain modulation functions to the two populations. Importantly, no synaptic coupling was present between them. All neurons were simulated using the same EIF model described earlier. Hyperparameters for the GLM-Transformer were also the same as in the previous figure.

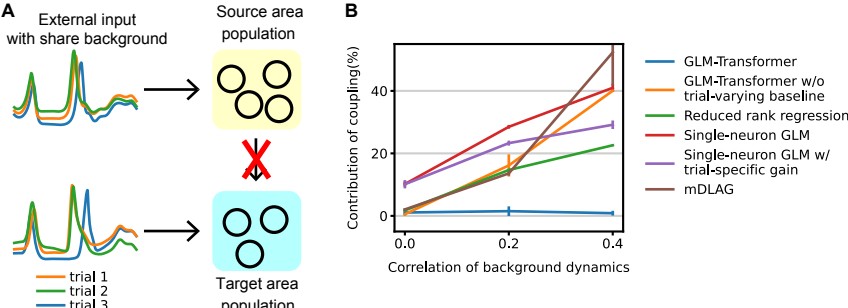

Figure 4: Robustness to shared background dynamics. (**A**) Schematic of the simulation setup: two neural populations receive correlated external input but have no direct synaptic connections between them. (**B**) Percentage of variance in predicted firing rates attributed to coupling effects, as a function of background gain correlation. Although no true coupling exists, most baseline methods spuriously attribute correlated inputs to coupling. In contrast, GLM-Transformer maintains low estimates across conditions, due to its trial-varying individual-neuron dynamics component. The x-axis represents correlation in the underlying Gaussian process gain modulation, rather than the total input current.

We compared GLM-Transformer to several existing methods, including reduced-rank regression (RRR) [48], delayed latents across multiple groups (mDLAG) [47], and single-neuron GLMs (with and without trial-wise constant gain modulation) [6]; we also included a variant of GLM-Transformer without trial-varying individual-neuron dynamics. Figure 4 shows the proportion of variance in predicted firing rates attributed to coupling effects as the correlation in shared nonstationary gain increases. While all other methods incorrectly attributed shared nonstationary gain to coupling, the GLM-Transformer consistently maintained low estimates of spurious coupling across all conditions. Notably, the trial-varying individual-neuron dynamics component plays a critical role, as removing it from our model (see the orange curve in Figure 4) leads to overestimated coupling effects, similar to the other methods.

## 4.5  Identifying cross-area interactions in the Allen Institute Visual Coding dataset

We applied GLM-Transformer to the Allen Institute Visual Coding dataset, which contains simultaneous Neuropixels recordings from six visual areas in awake mice, including the primary visual cortex (V1), the lateral medial area (LM), and the anterolateral area (AL). During each trial, mice passively viewed visual stimuli including Gabors, drifting gratings, and natural movies [42]. The model was trained on all available data from ten animals, comprising 121,679 trials of 400 ms each. After hyperparameter tuning, we used two factors for individual-neuron dynamics, one factor for each coupling term, and chose $d_z = 4$ (see Appendix E for details).

GLM-Transformer identified strong coupling from V1 to LM and from V1 to AL (Supplementary Figure S4), consistent with the known visual hierarchy [28] and prior findings from the same dataset [46]. Figure 5 presents detailed results for LM's individual-neuron dynamics and the coupling from V1 to LM in one animal previously analyzed by Chen et al. [46]. As shown in Figure 5A, LM's individual-neuron dynamics exhibit substantial trial-to-trial variability. Notably, the neurons with the strongest coupling weights, both in V1 (sending) and in LM (receiving), substantially overlap with the "cross-pop" neurons identified by Chen et al. [46] (Figure 5D, neurons in red). Chen et al. showed that this subset of neurons plays a central role in cross-area interactions, as the population firing peak times in V1 and LM were highly correlated when computed using only these neurons. In contrast, we observe less alignment between these "cross-pop" neurons and the loading weights of LM's individual-neuron dynamics factors (Figure 5B), suggesting that the two terms capture different patterns.

To further examine the trial-wise latent $\mathbf{z}_r^{a,j}$, we visualized LM's latents using t-SNE (Figure 5E). The embedding shows clear organization with respect to locomotion and pupil diameter, a proxy for arousal. Since these behavioral variables are major sources of background dynamics across the brain [13, 14, 15, 16], this alignment suggests that the inferred trial-wise latent captures meaningful trial-to-trial variability.

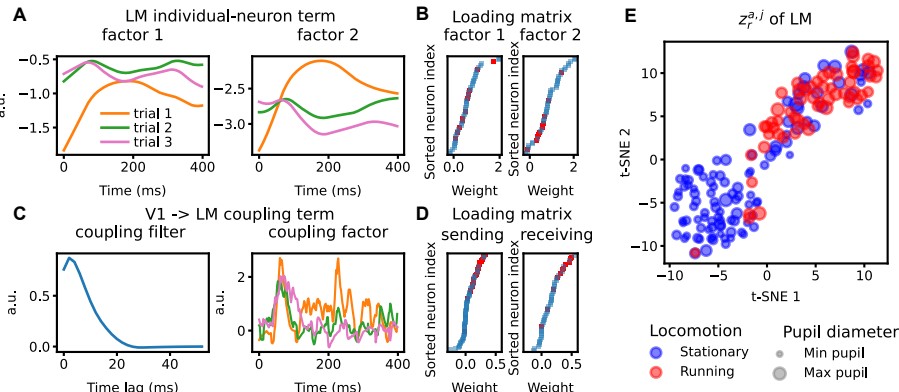

Figure 5: Individual-neuron dynamics and coupling effects inferred by GLM-Transformer in the Allen Institute Visual Coding dataset. (**A**) Trial-specific individual-neuron dynamics in LM are captured by two smooth latent factors, shown across three example trials. (**B**) Loading weights from the individual-neuron dynamics factors to LM neurons. Red squares mark "cross-pop" neurons previously identified by Chen et al. [46] as participating in cross-area coordination. (**C**) Coupling filter (left) from V1 to LM and the corresponding coupling factor for three example trials (right). We only have one coupling factor from an area to another. (**D**) Sending and receiving weights of the coupling factor from V1 to LM. Red squares again indicate the "cross-pop" neurons, which match the neurons with highest sending/receiving weight in GLM-Transformer. (**E**) t-SNE embedding of the trial-wise latent variable $\mathbf{z}_r^{a,j}$ of LM colored by behavioral state (locomotion: stationary vs. running) and sized by pupil diameter. The learned latents capture both locomotion and pupil diameter effectively.

The alignment between "cross-pop" neurons and those with large coupling weights, along with the organization of trial-wise latents by locomotion and arousal, provides indirect evidence for the reliability of GLM-Transformer's two key components: coupling and individual-neuron dynamics. These results suggest that GLM-Transformer can jointly capture interpretable cross-area interactions and rich trial-to-trial variability.

## 5  Discussion

The GLM-Transformer combines GLM-style coupling components with a flexible Transformer model for within-area dynamics. It thereby enables interpretable estimation of directed interactions across populations of spike trains in the presence of trial-varying dynamics, as observed in experimental data [13, 14, 15, 16]. The method is also scalable in practice, supporting joint training across multiple animals, sessions, and datasets, which can improve generalization and robustness.

Despite its advantages, the GLM-Transformer has several limitations that could be explored in future work. One issue is the dependence of the Transformer on training. Although our multi-stage training procedure mitigates some of the variability across alternative settings, and we observe consistent results across a broad range of hyperparameters, there are no theoretical guarantees on performance. It is easy to imagine a substantial effort devoted to training under a wide variety of circumstances based on the ever-increasing availability of multiple spike train data. In addition, GLM-Transformer is more data demanding than traditional GLMs, typically requiring over a thousand trials to converge, which is about an order of magnitude more data than standard GLMs. This limitation could be mitigated by initializing the model from a pretrained network when only a small dataset is available.

We have also made several simplifying assumptions. For example, our implementation assumes a fixed trial length, which limits applicability to datasets with variable temporal structures. A practical workaround for datasets with a few discrete trial lengths would be to learn separate decoders per trial type. A more general extension would involve replacing the spline-based decoder with a GPT-style architecture capable of handling variable-length sequences, which could broaden the scope of tasks addressable by GLM-Transformer. In addition, we currently use the same number of factors for the individual-neuron and coupling components, a choice that could be relaxed in future work.

Finally, a key scientific challenge is to develop methods for statistical inference about effects of interest in models like GLM-Transformer, where repeated retraining would be computationally expensive. Recent research on post-hoc debiasing and uncertainty-aware estimation [53, 54, 55] suggests these strategies might be made applicable to GLM-Transformer. Once rigorous inference about cross-area interaction effects becomes available, models following the general approach deployed in GLM-Transformer could become useful in many studies involving large-scale neural recordings.

## Code availability

GitHub repository with code to reproduce all experiments is available at: `https://github.com/Qi-Xin/GLM-Transformer`.

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

# A   Ablation Study

Table S1: Ablation study results. Values are test loss in terms of negative log-likelihood (lower is better), with standard error in parentheses.

| Model Variant | Dataset | | |
|---|---|---|---|
| | GLM Simulation | EIF Simulation | Allen Visual Coding |
| GLM-Transformer (Full) | $3.008 \times 10^7$ $(2.6 \times 10^4)$ | $1.513 \times 10^7$ $(4.9 \times 10^3)$ | $4.218 \times 10^7$ $(3.8 \times 10^4)$ |
| Bidirectional RNN encoder | $3.032 \times 10^7$ $(5.3 \times 10^4)$ | $1.515 \times 10^7$ $(1.0 \times 10^4)$ | $4.225 \times 10^7$ $(2.7 \times 10^4)$ |
| No trial-wise latent | $3.035 \times 10^7$ $(4.4 \times 10^4)$ | $1.521 \times 10^7$ $(7.3 \times 10^3)$ | $4.264 \times 10^7$ $(3.3 \times 10^4)$ |
| No coupling term | $3.065 \times 10^7$ $(1.2 \times 10^5)$ | $1.537 \times 10^7$ $(7.3 \times 10^3)$ | $4.263 \times 10^7$ $(2.6 \times 10^4)$ |
| No post-spike history | $3.109 \times 10^7$ $(4.1 \times 10^4)$ | $1.603 \times 10^7$ $(2.8 \times 10^4)$ | $4.294 \times 10^7$ $(1.0 \times 10^4)$ |

To assess the contribution of key components of GLM-Transformer, we conducted a series of ablation experiments, with results summarized in Table S1. Each variant removes or alters one component of the full model, and we report the resulting test negative log-likelihood (NLL) on three datasets: a synthetic GLM simulation, an EIF simulation, and the Allen Institute Visual Coding dataset. In all experiments, we use the mean of the approximate posterior as the latent without sampling.

Replacing the Transformer encoder with a bidirectional RNN yields slightly higher test loss on the synthetic dataset, with minimal differences on the EIF and real datasets. This suggests that while the Transformer may better capture certain trial-specific features, both architectures can perform comparably in practice. Removing the trial-wise latent variable entirely leads to a further degradation in performance across all datasets, confirming the importance of capturing trial-to-trial variability in the individual-neuron dynamics component. Omitting the coupling term or the post-spike history filter significantly worsens model fit.

Overall, the ablation results validate the design of GLM-Transformer: each component contributes meaningfully to the model's performance, and the full model consistently achieves the best results across all settings.

# B   Supplementary figures

## B.1   Non-identifiability example

Supplementary Figure S1 illustrates a case of non-identifiability when regularization is removed and the individual-neuron dynamics component is made overly flexible. In this example, the model is configured with a larger number of B-spline basis functions (increased from 10 to 50), no smoothness penalty (regularization weight reduced from 1000 to 0), a high-dimensional trial-wise latent variable $\mathbf{z}_r^{a,j}$ (dimension increased from 4 to 64), and more latent factors for the individual-neuron dynamics term (increased from 2 to 4). As a result, the individual-neuron dynamics component absorbs temporal structure that should instead be attributed to coupling.

While the total predicted log firing rate closely matches the ground truth, the decomposition into individual-neuron dynamics and coupling is incorrect. This highlights the necessity of adopting modeling assumptions that make the two effects distinguishable. In our case, we impose smoothness on the individual-neuron dynamics.

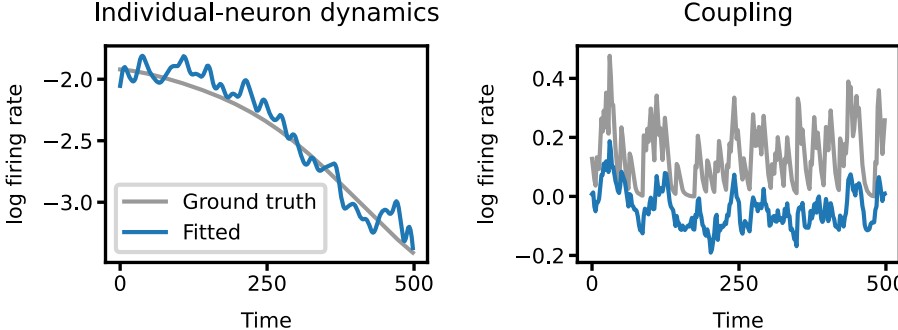

Supplementary Figure S1: Non-identifiability example from the GLM simulation dataset. Left: Ground truth (gray) and fitted (blue) individual-neuron dynamics. Right: Ground truth (gray) and fitted (blue) coupling term. Although the predicted firing rate is somewhat correct, the decomposition is incorrect: the overly flexible individual-neuron dynamics absorbs the coupling component.

## B.2 Post-spike filter recovery

All neurons in the GLM simulation share the same ground truth self-history filter, which reflects a refractory period. Figure S2 shows that GLM-Transformer recovers the shape and timing of this filter across neurons.

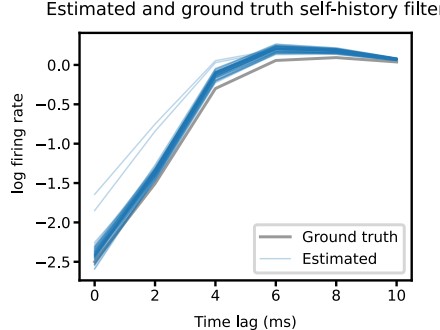

Supplementary Figure S2: Recovery of the shared post-spike filter in the GLM simulation. The estimated filter (blue) matches the ground truth (gray).

## B.3 EIF simulation with four neural populations

We conducted an additional EIF simulation involving four neural populations to evaluate the scalability of GLM-Transformer. The ground-truth connectivity was designed such that population 1 projected to populations 2 and 4, and population 2 projected to population 3. GLM-Transformer successfully recovered the true directed interactions, demonstrating its ability to generalize to more than two neural populations.

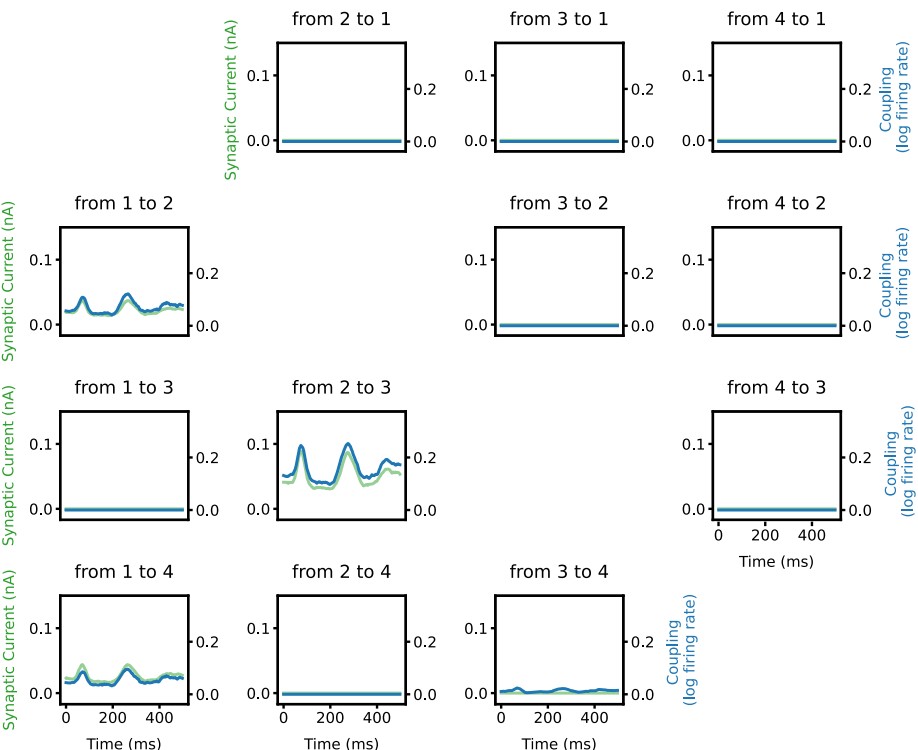

Supplementary Figure S3: EIF simulation with four neural populations, each consisting of 50 neurons. Each panel shows the estimated coupling filters (blue) and corresponding ground-truth synaptic currents (green) for all pairs of source and target populations. GLM-Transformer successfully identifies the true connectivity structure (1→2, 2→3, 1→4).

## B.4 Outputs of the six area in Allen Institute Visual Coding dataset

We applied GLM-Transformer to six visual cortical areas in the Allen Institute Visual Coding dataset (V1, LM, AL, RL, AM, PM). Each subplot shows the baseline and estimated coupling effects from one source area to a target area. Consistent with known anatomical hierarchies, the model identified strong feedforward connections from V1 to higher visual areas, especially V1→LM and V1→AL. Therefore, we focus on V1→LM in Figure 5. These results further suggest that GLM-Transformer can extract interpretable and biologically meaningful connectivity patterns from large-scale neural recordings.

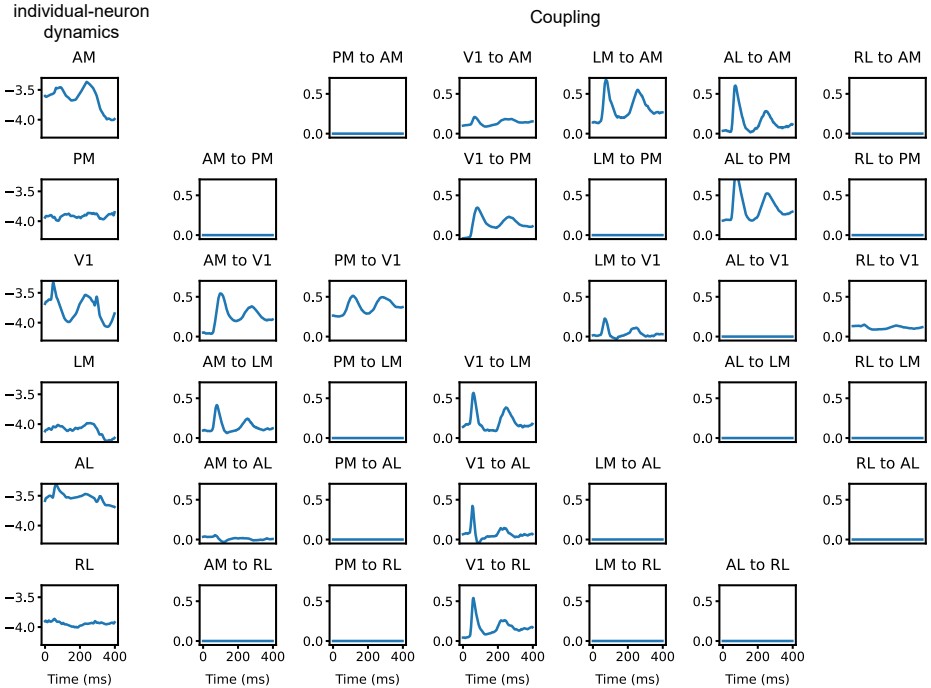

Supplementary Figure S4: Outputs of the six areas in the Allen Institute Visual Coding dataset. Each row shows the individual-neuron dynamics (left column) and estimated coupling effects (remaining columns) for one target area. Strong coupling is observed from V1 to LM and from V1 to AL, consistent with the known feedforward hierarchy of the mouse visual cortex.

## C  Regularization terms

We use two forms of regularization to improve identifiability and interpretability.

**Smoothing penalty.**    To encourage smooth temporal structure in the individual-neuron dynamics, we apply an $\ell_2$ penalty on the second-order finite differences of each latent factor. Specifically, let $f(t) \in \mathbb{R}^T$ denote a time-varying factor evaluated at $T$ discrete time bins. The smoothing penalty is:

$$\gamma_{\text{smooth}} \cdot \frac{1}{T} \sum_{t=2}^{T-1} \left( f(t+1) - 2f(t) + f(t-1) \right)^2,$$

where $\gamma_{\text{smooth}}$ is a tunable regularization weight. This penalizes curvature in the factor trajectory, promoting smoothness over time.

**Sparsity penalty.**    To encourage sparsity in the coupling structure, we apply an $\ell_1$ penalty to the sending and receiving weights of the coupling factors.

## D  Shared-across-animals and animal-specific components

To encourage the model to learn fundamental features that generalize across animals, we promote parameter sharing wherever feasible. For the individual-neuron dynamics component, the first and last processing steps are kept animal-specific, as they directly operate on neurons, and each animal involves a different set of neurons. However, all intermediate steps—including the Transformer encoder and MLP decoder—are shared across animals and brain areas.

In contrast, the coupling component involves only two processing steps, and none of its parameters are shared across animals. This design choice allows the model to capture animal-specific variations in functional interactions across brain areas.

# E   Hyperparameter tuning

For the Allen Institute Visual Coding dataset, we performed hyperparameter tuning for GLM-Transformer using Bayesian optimization with the `hyperopt` library and the Tree-structured Parzen Estimator (TPE) algorithm. The objective was to minimize the test negative log-likelihood.

Each trial followed the four-stage training procedure described in the main text, and evaluation used the mean of the approximate posterior (i.e., without sampling). Due to computational constraints, hyperparameter optimization was conducted on a subset of the data consisting of two animals (IDs: 757216464 and 715093703), and the resulting optimal configuration was used for all real-data experiments.

**Search space.**   The following parameters were tuned:

- **B-spline basis:** number of basis functions in $\{10, 30, 50\}$
- **Transformer encoder:** number of layers in $\{1, 2, 4\}$, hidden size $d_{\text{model}}$ in $\{128, 256, 512\}$, feedforward dimension in $\{256, 512, 1024\}$, dropout in $\{0.0, 0.2, 0.4\}$, number of attention heads in $\{1, 2\}$, and token time bin length (ms) in $\{10, 20\}$
- **Trial-wise latent variable $\mathbf{z}_r^{a,j}$:** dimensionality $d_{\mathbf{z}}$ in $\{2, 4, 8\}$
- **Decoder:** number of latent factors in $\{1, 2, 4\}$ and MLP hidden layer size in $\{d_{\mathbf{z}}, 2d_{\mathbf{z}}, 4d_{\mathbf{z}}\}$
- **Coupling:** number of basis functions in $\{3, 5\}$, window length (maximum delay allowed) in $\{30, 50, 70\}$, coupling subspace dimensionality in $\{1, 2\}$
- **Penalties:** spline smoothing in $\{100, 1000, 10000\}$, coupling subgroup penalty in $\{1\text{e-}5, 1\text{e-}4, 1\text{e-}3, 1\text{e-}2\}$
- **Optimization:** Adam optimizer with learning rates for each component in $\{1\text{e-}4, 1\text{e-}3, 1\text{e-}2\}$, batch size in $\{32, 64, 128\}$

**Final selected hyperparameters.**   The best-performing configuration was:

- **B-spline basis:** 10 basis functions
- **Transformer encoder:** 1 layer, $d_{\text{model}} = 128$, feedforward dimension = 512, dropout = 0.0, 1 attention head, token time bin length 20 ms
- **Trial-wise latent variable $\mathbf{z}_r^{a,j}$:** $d_{\mathbf{z}} = 4$
- **Decoder:** 2 individual-neuron dynamics factors, MLP hidden layer size = 4
- **Coupling:** 3 basis functions, window length (maximum delay allowed) = 50, 1 subspace dimension
- **Self-history filters:** 3 basis functions, window length (maximum delay allowed) = 10
- **Penalties:** spline smoothing = 1000, coupling subgroup penalty = 1e-5
- **Optimization:** Adam optimizer with transformer encoder learning rate = 1e-4, decoder and coupling learning rates = 1e-2, and batch size = 64.

For the GLM simulation dataset, we use the same set of hyperparameters as those selected for the Allen Institute data. For the EIF simulation dataset, we use a smaller spline smoothing penalty (= 100) and a shorter coupling effect window (35 ms); all other hyperparameters remain the same.

We also explored the robustness of the model by jittering a few hyperparameters around the final selected values. The results remained consistent: the inputs from V1 to LM and from V1 to AL were almost always the strongest coupling effects.

## F  Exponential integrate-and-fire (EIF) neuron model

We used the exponential integrate-and-fire (EIF) neuron model [49, 50, 51] to generate synthetic datasets. Unlike the simpler leaky integrate-and-fire (LIF) model, the EIF model more accurately captures the exponential rise in membrane potential near the firing threshold and is considered more biologically realistic [49]. Our simulated spiking network consists of two neural populations, source and target, each containing 50 neurons. The membrane potential $V_n(t)$ of neuron $n$ evolves according to:

$$C_m \frac{dV_n}{dt} = -g_L(V_n - E_L) + g_L \Delta_T e^{(V_n - V_T)/\Delta_T} + I_n^{\text{syn}}(t) + I_n^{\text{noise}}(t) + I_n^{\text{ext}}(t). \tag{7}$$

A spike is emitted when $V_n(t)$ reaches the threshold $V_{\text{th}}$. The neuron then enters a refractory period $\tau_{\text{ref}}$, after which its membrane potential is reset to $V_{\text{re}}$. We use the same parameters as in Huang et al. [51]: $\tau_m = C_m/g_L = 15\,\text{ms}$, $E_L = -60\,\text{mV}$, $V_T = -50\,\text{mV}$, $V_{\text{th}} = -10\,\text{mV}$, $\Delta_T = 2\,\text{mV}$, $V_{\text{re}} = -65\,\text{mV}$, and $\tau_{\text{ref}} = 1\,\text{ms}$.

The input terms $I_n^{\text{syn}}(t)$, $I_n^{\text{noise}}(t)$, and $I_n^{\text{ext}}(t)$ represent synaptic input, background noise, and external stimulus, respectively. $I_n^{\text{noise}}(t)$ is modeled as Gaussian white noise with amplitude selected to yield a baseline firing rate of 10 Hz in the absence of synaptic or external inputs. The synaptic input current is given by:

$$\frac{I_n^{\text{syn}}(t)}{C_m} = \sum_{m=1}^{N} J_{n,m} \sum_k \eta(t - t_k^m), \tag{8}$$

where $J_{n,m}$ is the synaptic weight from neuron $m$ to neuron $n$, and $t_k^m$ denotes the $k$th spike time of neuron $m$. The postsynaptic current kernel $\eta(t)$ is defined as:

$$\eta(t) = \frac{1}{\tau_d - \tau_r} \begin{cases} e^{-t/\tau_d} - e^{-t/\tau_r}, & t \geq 0, \\ 0, & t < 0, \end{cases} \tag{9}$$

with $\tau_d = 5\,\text{ms}$ and $\tau_r = 1\,\text{ms}$. Synaptic weights are sampled from a log-normal distribution [52]. In Figure 3, the mean synaptic weight from 10 source neurons to 10 target neurons is 0.2. Recurrent connections within each population have a mean weight of 0.01.

The external input $I_n^{\text{ext}}(t)$ consists of two components. The first mimics visual stimuli from the Allen Institute Visual Coding dataset using a dual-peaked temporal profile with peaks at approximately 50 ms and 250 ms after stimulus onset [46]. The second component introduces trial-to-trial variability through gain modulation. A smooth gain factor is drawn per trial from a Gaussian process (with a time constant of 300 ms and amplitude 0.05) and projected to each neuron using a loading vector whose entries are independently sampled from Uniform$(0.5, 1.0)$. In Figure 3, the Gaussian process factors of the two populations are uncorrelated, while they are correlated in Figure 4.

Simulations were performed with a 0.2 ms time step, and spike trains were then binned into 2 ms intervals for use in GLM-Transformer and other models. A total of 2,000 trials were simulated.

## G  Animals used in the Allen Institute Visual Coding dataset

All real-data experiments (e.g., Figure 5 and Supplementary Figure S4) were conducted using Neuropixels recordings from ten animals in the Allen Institute Visual Coding dataset. The animal IDs are: 757216464, 798911424, 715093703, 719161530, 721123822, 737581020, 739448407, 742951821, 743475441, and 744228101.

We used all available trial types. Trials shorter than 400 ms were concatenated with the following trial, since these stimuli (e.g., Gabors, flashes) were presented without intervening blackout periods. For trials longer than 400 ms (e.g., drifting gratings, which last 500 ms), we used only the first 400 ms. For stimulus types that span long continuous durations (e.g., natural movies), we divided the full sequence (e.g., 10 minutes) into non-overlapping 400 ms segments.

