# OpenReview forum: "Identifying interactions across brain areas while accounting for individual-neuron dynamics with a Transformer-based variational autoencoder"
_NeurIPS.cc/2025/Conference — NeurIPS 2025 poster_

### Official Review · Reviewer_wbH6 · 2025-06-01

**Clarity:** 3
**Significance:** 2
**Originality:** 2
**Rating:** 4
**Confidence:** 4

**Summary:**

This paper proposes a GLM-Transformer for multi-population and multi-trial neural spike data. The GLM-Transformer generalizes the widely used GLM with a Transformer-based VAE, which decomposes neural activity into three components: individual-neuron dynamics, self-history effects and cross-population coupling. This decomposition makes the model flexible but keeps the interpretability as in GLM. After validating by simulations, they further applied it to the Allend Institute Visual Coding dataset and identified feed forward pathways consistent with established visual hierarchies.

**Questions:**

The paper is clearly written. Besides the question in the weakness section, I don’t have other major questions.

**Ethical Concerns:**

["NO or VERY MINOR ethics concerns only"]

**Final Justification:**

I am between borderline accept & reject, as reasons provided in the response. Will let AC make final decision.

**Limitations:**

The paper has discussed the limitations in the Discussion section: dependence of the Transformer on training, the assumption and inference.

**Paper Formatting Concerns:**

No formatting concerns

**Quality:**

2

**Strengths And Weaknesses:**

**Strengths**:
- The work is well-motivated by challenges in analyzing real neural spike train data.
- The writing is clear, structured, and easy to follow.
- The model extends the GLM using a Transformer-based VAE in latent space, enhancing flexibility while preserving interpretability through a biologically meaningful decomposition.

**Weaknesses**:
- The GLM-Transformer combines GLM with Transformer-based VAE in a straightforward way, which may raise questions regarding novelty.
- Although the method focuses on the interpretability and inference. It would be better to compare it with other methods. Will simple GLM based method be enough to gain insights/ biological interpretation?

---

> ### Author Rebuttal · Authors · 2025-07-31
>
> We thank reviewer wbH6 for their helpful review and for recognizing our work is "well-motivated by challenges in analyzing real neural spike train data". We address your questions below.
>
> > Weakness) The GLM-Transformer combines GLM with Transformer-based VAE in a straightforward way, which may raise questions regarding novelty
>
> We appreciate this comment and would like to clarify our perspective. We believe novel methods often build on existing techniques. The GLM-Transformer (a) decomposes coupling identification from background dynamics and (b) takes advantage of the Transformer to do what it is good at: flexibly fitting complicated temporal processes where interpretation is not needed. This is a novel conception and we spent considerable time and effort to demonstrate its viability.
>
> > Weakness) Although the method focuses on the interpretability and inference. It would be better to compare it with other methods. Will simple GLM based method be enough to gain insights/ biological interpretation
>
> We fully agree about the importance of comparing to simpler GLM-based baselines and we did include such a comparison. As shown in Figure 3, we directly compared GLM-Transformer against simple GLMs and other alternatives, in the presence of shared background dynamics. These comparisons demonstrate that while standard GLMs can provide interpretable components, they lead to biased coupling when background dynamics are present. GLM-Transformer remains robust in this setting by modeling trial-varying individual-neuron dynamics explicitly, and we discuss this result in the main text.

---

> > ### Comment · Reviewer_wbH6 · 2025-08-06
> >
> > Thank the authors for their response. The comparisons to simple GLMs and other methods look good, but I'm still concern about the novelty issue. Therefore, I tend to keep my initial evaluation.

---

### Official Review · Reviewer_pEdw · 2025-06-17

**Clarity:** 2
**Significance:** 3
**Originality:** 2
**Rating:** 4
**Confidence:** 4

**Summary:**

This paper introduces GLM-Transformer, a hybrid model that combines a generalized linear model (GLM) for interpretable cross-population coupling with a Transformer-based variational autoencoder to capture individual-neuron dynamics and trial-to-trial variability. The authors estimate a low-dimensional latent vector per trial and population,m which maps onto B-spline-based background firing rate trajectories. This is combined with self-history and coupling terms that are modeled via low-rank temporal filters. The model is trained in four stages, iteratively increasing complexity, with an ablation study showing that every component meaningfully improves performance. The authors validate their approch on synthetic, mechanistic, and real spike train data, where it recovers known interaction structure.

**Questions:**

- assumption of fixed trial length: I initially found it a bit hard to understand how time-dependence arises from the fixed trial latent factors via the splines. It seems the approach assumes the splines are defined over a fixed temporal support. Could you clarify how restrictive this assumption is, e.g., how would the model handle variable-length trials or generalization beyond the training interval? You mention the fixed length briefly in the Discussion (“we also were able to assume a fixed trial length...”), but this assumption appears central and could benefit from more explicit discussion directly  in the Methods section.

- You mention that “in all experiments, we use the mean of the approximate posterior as the latent without sampling.” Out of curiosity: how large is the latent variance in practice? What would happen if one trained the model deterministically without a VAE setup? This would help clarify whether the probabilistic treatment adds substantial value (could also be added to table S1 if time allows)

- Table S1:The absolute differences in test loss are very small (e.g., 0.0567 vs. 0.0568 for Transformer vs. RNN encoder). Are these differences practically meaningful? Do they persist across longer trials or different datasets?

- Encoder: What adjustments, if any, were necessary to adapt the Transformer to spike data? Did you mainly take the settings from the literature? Are there specific preprocessing or normalization steps for the token sequences? While the hyperparameter search for the Transformer is in the supplement, no other details are given as far as I could see.
Second, how does computational load compare to the RNN variant, particularly as trial length increases? What is the computational load vs. the RNN/runtime? Especially with respect to trial length?

**Ethical Concerns:**

["NO or VERY MINOR ethics concerns only"]

**Final Justification:**

As highlighted in my final comment, a few limitations remain. First, the Transformer encoder, despite its prominence in the title, yields virtually no performance gain over a bidirectional RNN in your ablation (Table S1), while the suggestion that it will outperform on other datasets feels a bit speculative. Second, the fixed-length-trial assumption is quite substantial, since in many real-world settings trials vary substantially. Finally, as mentioned by other referees, the methodological novelty beyond the low-rank GLM integration feels incremental. I usually don't like raising the novelty claim, since it is hard to argue against, and the authors rightly acknowledge that they're building on considerable prior work, but for that reason I’d expect the other parts of the paper (e.g., more compelling empirical gains or broader applicability) to compensate.

That being said, overall, the GLM-Transformer represents an interesting integration and yields solid empirical results, but these remaining concerns keep me on the fence. I will therefore retain my current borderline-accept rating.

**Limitations:**

Clear limitations section.

**Paper Formatting Concerns:**

No.

**Quality:**

3

**Strengths And Weaknesses:**

*Strengths*:

This is a well-written and logically structured paper that presents a principled approach to disentangling background activity and cross-population coupling in neural spike train data. The introduction is strong, and the method is clearly motivated. The combination of a GLM framework with a (Transformer-based) variational autoencoder is sensible and timely, and the authors have done a good job of situating their work within the relevant literature on spike train modeling. The four-stage training procedure and the corresponding ablation study in Table S1 help clarify the contributions of individual components. The results section is well-organized.

*Weaknesses*:

Modeling assumptions: While much of this is unavoidable when combining multiple components into sophisticated architectures, there are some points in the methods section where modeling assumptions or design decisions are introduced without enough justification. In particular:
- "to avoid this, we make the seemingly reasonable assumption that coupling
effects vary more rapidly than the background fluctuations arising from changes in behavior or endogenous state". It’s fair that we need to introduce some inductive biases to assure identifiability, but this assumption is not very intuitive to me. In practice, behavioral or arousal-related state changes (e.g., due to attention or movement) can also occur on fast time scales? What would be a scenario in which this breaks down?

- “We use 10 or 20 ms time bins as tokens for the Transformer (which are wider than the 2-ms bins used elsewhere in the model)” Why? I understand that this may be driven by attention scaling (e.g., quadratic cost), but an explicit justification would be useful.
-“Throughout the paper, we use a bin size of 2 ms and rarely observe
more than one spike per bin” Have you considered alternative count models (e.g., zero-inflated Poisson or negative binomial)?
- The posterior over trial-wise latent variables assumes a diagonal covariance matrix. While this is common, some discussion of whether this is a modeling assumption or a practical constraint (e.g., for stability or training speed) could be helpful.

Clarity of contribution:
While the method is well-motivated, I found the paper could more clearly articulate what is actually novel in terms of methodology. Many of the model components, e.g., trial-wise latent variables, B-spline decoders (as in Deep Random Splines [36]), smooth temporal filters via raised cosine bases [6], and Transformer encoders adapted to spike data [31], are taken from prior work. That's entirely valid, and combining these elements in a novel  architecture can itself be a contribution. However, a short summary paragraph at the end of the Methods section highlighting which components are standard, which are adapted, and which are new would be really helpful.

Framing and overall contribution:
Relating to the clarity of contribution, one point that could also benefit from clarification is the overall goal of the paper. The title emphasizes “identifying interactions across brain areas”, which would suggest that the coupling terms are the object of interest. However, much of the model’s complexity and interpretability comes from the other components (the individualized response functions, trial-specific latent variables and self-history terms). It’s not always clear how central they are to the paper’s scientific aims. Are the latents z meant to be interpreted (e.g., in relation to behavior, as in Figure 5E), or are they primarily there to recover accurate coupling terms? A clearer statement early in the paper, and perhaps reiterated at the end of the Methods section, about what is meant to be interpreted versus abstracted away would be helpful.

Suggestions:
The related work section is concise and well-organized but somewhat narrow in scope. It focuses primarily on two points: (1) that Deep Random Splines (DRS) uniquely capture trial-level variability, and (2) that prior models do not provide interpretable population-level coupling. However, trial-specific modeling is an increasingly active area, with several recent approaches proposing hierarchical or meta-dynamical structures that extract trial-level latents in a dynamics context, e.g.
Meta-Dynamical State Space Models for Integrative Neural Data Analysis (Vermani et al, ICLR 2025)
Learning Interpretable Hierarchical Dynamical Systems Models from Time Series Data (Brenner et al., ICLR 2025)
Generalizing to New Physical Systems via Context-Informed Dynamics Model (Kirchmayer et al., ICML 2022).

Also, the claim that prior latent space models "do not provide direct interpretation" of cross-population effects could be clarified: why can such interpretability not also be achieved through structured latent recurrences or constrained connectivity (e.g., low-rank or sparse matrices). Finally, some recent work also focuses on directly inferring latent factors from spike trains that may also be worth referencing: e.g.
Latent Diffusion for Neural Spiking Data (Kapoor et al., NeurIPS 2024)
Integrating Multimodal Data for Joint Generative Modeling of Complex Dynamics (Brenner et al., ICML 2024)

---

> ### Author Rebuttal · Authors · 2025-07-31
>
> We thank Reviewer pEdw for the detailed and constructive review and for describing our work as “sensible and timely.” We greatly appreciate that you read deeply with the main text and supplementary materials. Your feedback has already helped us identify several ways to improve the work. We address your comments below, and ***we will revise the manuscript accordingly***.
>
> > Weakness 1) Modeling assumptions: assumption to assure identifiability
>
> This is good criticithsm. We overstated that point, and we changed "we made a seamingly reasonable assumption" to "we made a strong assumption" in our revision.
>
> Our reasoning is based on the fact that slower fluctuations in neural activity tend to have higher power in the spectrum [8]. By regularizing the individual-neuron dynamics to be smoother, we are effectively adjusting for background fluctuations that dominate lower-frequency bands.
>
> Also, in the Allen Institute Visual Coding dataset, we believe the major sources of trial-to-trial background fluctuation are behavioral factors such as locomotion, arousal, and reaction time. These tend to evolve on the scale of hundreds of milliseconds, whereas cross-area interactions typically occur at much faster time scales [9]. So, we believe our assumption holds in this setting and in many similar experimental paradigms. That said, we acknowledge that in other situations (e.g., fast-changing internal states), the assumption may be violated.
>
> Lastly, we want to emphasize that we view functional coupling as capturing conditional dependencies, not causality. Even if some fast-changing unobserved confounders exist, the estimated coupling that is conditioned on the slow-changing background component, can still reflect useful and interpretable patterns of interaction across areas.
>
> > Weakness 1) Modeling assumptions: explicit justification for 10ms or 20ms time bin
>
> Another reviewer also brought this up. We tried shorter or longer time bin widths; the performance (log likelihood) drops slightly if using time bins bigger than 20ms or smaller than 10ms. The model is relatively insensitive within the 10–20 ms range, and we found no substantive differences between these settings. A possible explanation is that larger time bins lose temporal information, while very small bins require more data to train and may not carry too much additional information.
>
> Below is a table showing the relative negative log-likelihood (NLL) of the model across different time bin widths for each dataset. Values are offset by the best (minimum) NLL for each dataset, so lower is better, and 0 indicates the best performance.
>
> Although the 20 ms bin width appears to perform better than 10 ms in this table, the results here were obtained on a smaller dataset ( 2000, 2000, and 5915 trials for the three datasets, respectively) due to limited time during the rebuttal phase. In contrast, our formal hyperparameter tuning (performed on a larger real dataset)  selected 10 ms as the optimal bin width, which is what we used in the final model.
>
> | Time bin width (ms)  | NLL on GLM synthetic dataset  | NLL on EIF synthetic dataset | NLL on Allen Institute dataset
> | ------------- |:-------------:| ------------- |:-------------:|
> | 2      | 2.43e+04     | 3.39e+04      | 6.39e+04     |
> | 5      | 1.09e+04     | 3.07e+04      | 6.30e+04     |
> | 10      | 1.43e+02     | 9.56e+01      | 1.42e+04     |
> | 20      | 0     | 0      | 0     |
> | 40      | 2.21e+03     | 6.03e+04      | 1.35e+04     |
>
>
> > Weakness 1) Modeling assumptions: posterior assumes a diagonal covariance matrix
>
> This is primarily a practical constraint. Modeling a full covariance matrix would require predicting and regularizing a positive semi-definite matrix with quadratic complexity, which would substantially increase model and training complexity. While possible, we found diagonal posteriors sufficiently expressive in practice, and the added cost was not justified for our current goals.
>
> > Weakness 2) Clarity of contribution
>
> We will absolutely add a summary paragraph at the end of the Methods section to clearly delineate which model components are standard, adapted, or novel. Due to length limitations in this rebuttal, we outline the key points below:
>
> - The individual-neuron dynamics component adapts ideas from Deep Random Splines (DRS) [10] and Neural Data Transformer (NDT) [11]: we use DRS’s latent-per-trial structure and B-spline decoder, NDT’s Transformer architecture, and dimension reduction to couple multiple neurons through shared latent structure.
> - The low-rank GLM-style coupling component is, to our knowledge, a novel integration. While inspired by reduced-rank regression, we are not aware of prior work combining it with spike-based GLMs to model directed interactions.
> - The post-spike history term is standard in GLM literature.
> - The overall design, which explicitly separates trial-varying background dynamics from coupling effects by leveraging the complementary strengths of GLMs and deep learning models, is also novel and central to our contribution.
>
> > Weakness 3) Framing and overall contribution
>
> Thank you for pointing this out. Our primary scientific goal is to estimate cross-area coupling. The individual-neuron latent component is treated as a nuisance term, which is important for isolating coupling, but not the focus of interpretation. In Figure 5E, we show that the latent aligns with behavior to help validate that it is capturing meaningful trial-to-trial structure. We will revise both the Introduction and Methods to better clarify this point.
>
> > Weakness 4) related work section
>
> Thank you for the helpful suggestions! We will incorporate the referenced recent works on trial-wise latent modeling.
>
> > Weakness 5) prior latent space models
>
> Yes, we agree that interpretability can also be achieved through structured latent connectivity, and there are many valuable models that take this approach. We think these are all important models for understanding population dynamics.
>
> That said, our GLM-style coupling term captures interactions in a different way. Rather than modeling communication from latent variables to latent variables, it directly estimates the effect of a spike in one neuron on the firing rate of another neuron at different time lags. This more explicit spike-to-rate formulation offers interpretable coupling filters that are especially intepretable for studying directed interactions between brain areas. We see this as a complementary perspective to latent-to-latent coupling approaches.
>
> > Q1) assumption of fixed trial length
>
> We agree this is a central assumption and currently a limitation of our model. For datasets with a few discrete trial lengths, one workaround would be to learn separate decoders per trial length. A more general and exciting direction would be to use a GPT-style decoder that supports variable-length outputs.
>
>
> > Q2) latent variance
>
> Thank you for catching this. In our ablation experiments, we disabled sampling to reduce variance in training and isolate the impact of other components. However, when sampling is turned off entirely, the posterior variance is no longer trained. In the real-data experiments, we do include sampling. The average standard deviation of the latent posterior is 0.57, averaged across dimensions and trials, indicating meaningful variability is captured.
>
> > Q3) Table S1
>
> We apologize for the confusion and thank you for pointing this out. The values in Table S1 are the negative log likelihood averaged over hundreds of thousands of time bins, so they are much smaller than the actual total negative log likelihood, which would require multiplying these numbers by the total number of time bins. This averaging compresses the scale and can make differences look deceptively small. We will change all the metrics to total negative log likelihood in the revision.
>
> As for the Transformer vs. RNN encoder comparison, the difference in negative log likelihood on the Allen dataset is within the confidence interval, so we don’t claim a strong advantage in this setting. However, we believe the benefit of the Transformer may become clearer in settings with longer trials or more complex neural dynamics, where its ability to model long-range dependencies might provide more value.
>
> > Q4) Encoder details
>
> We did not apply any special preprocessing or normalization to the token sequences; we largely followed prior practices from Neural Data Transformer [11]. In terms of runtime, we did not observe a major difference between the Transformer and the RNN encoder in our experiments. This is likely because the number of time bins (i.e., tokens) per trial is relatively small. In practice, the main bottleneck was converting sparse spike times to dense spike trains during data loading, which dominated the overall compute time and reduced the impact of encoder choice.
>
>
> ### References
>
> [8] Kass R E, Bong H, Olarinre M, et al. Identification of interacting neural populations: methods and statistical considerations[J]. Journal of Neurophysiology, 2023, 130(3): 475-496.
>
> [9] Chen Y, Douglas H, Medina B J, et al. Population burst propagation across interacting areas of the brain[J]. Journal of Neurophysiology, 2022, 128(6): 1578-1592.
>
> [10] Loaiza-Ganem G, Perkins S, Schroeder K, et al. Deep random splines for point process intensity estimation of neural population data[J]. Advances in Neural Information Processing Systems, 2019, 32.
>
> [11] Ye J, Pandarinath C. Representation learning for neural population activity with neural data transformers[J]. arXiv preprint arXiv:2108.01210, 2021.

---

> > ### Comment · Reviewer_pEdw · 2025-08-05
> >
> > Thank you for the thorough revision, which has adressed many of my points, also regarding the scope and purpose of the paper.
> >
> > That said, a few limitations remain. First, the Transformer encoder, despite its prominence in the title, yields virtually no performance gain over a bidirectional RNN in your ablation (Table S1), while the suggestion that it will outperform on other datasets feels a bit speculative. Second, the fixed-length-trial assumption is quite substantial, since in many real-world settings trials vary substantially. Finally, as mentioned by other referees, the methodological novelty beyond the low-rank GLM integration feels incremental. I usually don't like raising the novelty claim, since it is hard to argue against, and you rightly acknowledge that you’re building on considerable prior work, but for that reason I’d expect the other parts of the paper (e.g., more compelling empirical gains or broader applicability) to compensate.
> >
> > That being said, overall, the GLM-Transformer represents an interesting integration and yields solid empirical results, but these remaining concerns keep me on the fence. I will therefore retain my current borderline-accept rating.

---

> > > ### Author Response · Authors · 2025-08-05
> > >
> > > We deeply appreciate your detailed and thoughtful feedback! We found your suggestions really insightful and will revise the manuscript to better reflect the scope, limitations, and framing of the work.
> > >
> > > Thank you again for your careful review and engagement throughout the process!

---

### Official Review · Reviewer_BKwJ · 2025-06-30

**Clarity:** 2
**Significance:** 2
**Originality:** 3
**Rating:** 4
**Confidence:** 3

**Summary:**

This paper introduces GLM-Transformer, a new method for identifying how different brain areas communicate with each other using neural spike data. The main challenge is that neural activity varies significantly from trial to trial due to factors like behavioral changes, which can mask true interactions between brain regions. The authors' approach combines a Transformer-based neural network to capture these trial-to-trial variations in individual neuron activity with a generalized linear model framework to identify directed connections between brain areas. They tested their method on both synthetic data and real recordings from mouse visual cortex, showing that it can accurately recover known connections while remaining robust to confounding background activity that trips up other methods. When applied to the Allen Institute Visual Coding dataset, GLM-Transformer successfully identified feedforward pathways from V1 to higher visual areas, consistent with established brain anatomy. The method offers a way to better understand brain connectivity by separating true inter-area communication from within-area variability.

**Questions:**

1. Could you explain the mechanistic role of trial-varying individual-neuron dynamics in achieving robustness to shared background fluctuations?

2. Could you design a statistical framework to assess the significance of detected coupling effects and provide confidence intervals for coupling strength estimates?

3. Could you design a simulation experiment to evaluate the model's performance and computational scalability across varying numbers of brain areas (>2)?

**Ethical Concerns:**

["NO or VERY MINOR ethics concerns only"]

**Final Justification:**

My concerns have been resolved based on the new results and the authors' clarifications.

**Limitations:**

The authors discussed the limitations in Section 5, and I agree with their statements.

**Quality:**

3

**Strengths And Weaknesses:**

**Strengths**:

1. This paper addresses a challenge in computational neuroscience by combining the interpretability of GLMs with the flexibility of Transformer-based VAEs. This hybrid design is creative, allowing the model to capture complex trial-to-trial dynamics while maintaining interpretable cross-area coupling terms.

**Weaknesses**:

1. All experiments are conducted on relatively small-scale systems (2-3 brain areas), with Figure 5 demonstrating interactions only between V1 and LM. This limited scope raises serious questions about the method's scalability to realistic neuroscience applications, which typically involve dozens of brain areas and hundreds of neurons.

---

> ### Author Rebuttal · Authors · 2025-07-31
>
> We thank Reviewer BKwJ for recognizing that “this hybrid design is creative” and for the thoughtful feedback. We agree it will help strengthen the work. We address your questions below, and ***we will revise the manuscript to reflect these valuable suggestions***.
>
> > Weakness 1) All experiments are conducted on relatively small-scale systems (2-3 brain areas)
>
> We apologize for not making it clearer in the original submission. In our real dataset experiments, the model was applied to **six brain areas**: V1, LM, AL, RL, AM, and PM, each with many tens of neurons.
>
> The table below shows the **mean coupling contributions** (in log firing rate, averaged across trials, neurons, and time) between areas. The V1 -> LM coupling is the largest among well-sampled regions, consistent with the known feedforward hierarchy in mouse visual cortex. This is why we focused on V1 -> LM coupling in Figure 5. We will also add a supplementary figure to show this.
>
> |From ↓ To → | V1 | LM | AL | RL | AM | PM
> | ------------- |:-------------:|:-------------:|:-------------:| ------------- |:-------------:|:-------------:|
> | V1      | \     | 0.153      | 0.075     | 0.029      | -0.006      |-0.023     |
> | LM      | 0.006     | \      | 0.000     | 0.000      | 0.108      |0.000     |
> | AL      | 0.000     | 0.000      | \     | 0.000      | 0.036      |0.187     |
> | RL      | 0.024     | 0.000      | 0.000     | \      | 0.000      |0.000     |
> | AM      | 0.013     | 0.108     | 0.000     | 0.000      | \      |0.000     |
> | PM      | 0.026     | 0.000      | 0.000     | 0.000      | 0.000      |\     |
>
> We also added a new simulated dataset with **four populations** to further support model scalability. Please see our response to Q3 below.
>
>
>
> > Q1) mechanistic role of trial-varying individual-neuron dynamics in achieving robustness to shared background fluctuations
>
> Sorry we may not have explained this very well. In practice, each neuron in each trial exhibits background fluctuations, and they are often correlated across areas [4,5,6]. However, Without modeling these, a method might mistake shared background variability for inter-area coupling.
>
> Our trial-specific individual-neuron dynamics component explicitly models this variability, thereby isolating shared background dynamics from true cross-area interactions. This decomposition prevents the model from erroneously attributing correlated activity to coupling. We will clarify this point in the revised text.
>
> > Q2) design a statistical framework to assess the significance of detected coupling effects and provide confidence intervals for coupling strength estimates
>
> We agree that developing such inference procedures is a very important next step. While it is beyond the scope of the current work, we are actively exploring approaches inspired by recent work on post-hoc inference in deep learning, such as double/debiasing machine learning [7]. These ideas could be extended to estimate uncertainty around coupling effects in GLM-Transformer, and we view this as a promising direction for future work.
>
> > Q3) design a simulation experiment to evaluate the model's performance and computational scalability across varying numbers of brain areas (>2)
>
> We appreciate this suggestion and have conducted a new simulation experiment with **four neural populations** to directly address this point.
>
> We created a new EIF dataset with populations A, B, C, and D, each containing 50 neurons, to mimic the activity patterns of V1, LM, AL, and RL in the real dataset. The ground truth connectivity structure was designed as follows: A to B, B to C, and A to D.
>
> The table below shows the mean contributions between areas (in log firing rate, averaged across trials, neurons, and time bins). As shown, GLM Transformer successfully recovers the ground truth connections:
>
> |From ↓ To → | A | B | C | D |
> | ------------- |:-------------:|:-------------:|:-------------:| ------------- |
> | A     | \     | 0.219      | 0.000     | 0.108      |
> | B     | 0.000     | \      | 0.473     | 0.000      |
> | C     | 0.000     | 0.000      | \     | 0.000      |
> | D     | 0.000     | 0.000      | 0.000     | \      |
>
> This larger scale dataset demonstrates that GLM-Transformer can scale to more complex multi area networks, and we plan to replace the original two population simulation (Figure 2) with this updated result in the revised manuscript.
>
> ### References
>
> [4] Musall S, Kaufman M T, Juavinett A L, et al. Single-trial neural dynamics are dominated by richly varied movements[J]. Nature neuroscience, 2019, 22(10): 1677-1686.
>
> [5] Stringer C, Pachitariu M, Steinmetz N, et al. Spontaneous behaviors drive multidimensional, brainwide activity[J]. Science, 2019, 364(6437): eaav7893.
>
> [6] International Brain Laboratory, Benson B, Benson J, et al. A brain-wide map of neural activity during complex behaviour[J]. Biorxiv, 2023: 2023.07. 04.547681.
>
> [7] Kennedy E H. Semiparametric doubly robust targeted double machine learning: a review[J]. Handbook of statistical methods for precision medicine, 2024: 207-236.

---

> > ### Comment · Reviewer_BKwJ · 2025-08-04
> >
> > Thank you for the clarification and additional results. My concerns have been resolved, and I have updated my score accordingly.

---

> > > ### Author Response · Authors · 2025-08-05
> > >
> > > Thank you very much for taking the time to read our rebuttal and for considering the clarifications and additional results. We truly appreciate your thoughtful engagement with our work and we will revise the manuscript accordingly to incorporate the improvements discussed.

---

### Official Review · Reviewer_pXQU · 2025-07-03

**Clarity:** 4
**Significance:** 3
**Originality:** 3
**Rating:** 5
**Confidence:** 4

**Summary:**

The authors present a hybrid approach to modeling neural population dynamics that accounts for 1) individual neuronal latent dynamics, 2) cross-area coupling, 3) post-spike filters and spike history using a combination of GLMs and transformers. The authors outline the construction of their model and convincingly validate it on both real and simulated (i.e. ground truth) data.

**Questions:**

I have 3 questions:
- Why did the authors use a combination of splines, raised cosine basis functions, and “penalized roughness” to estimation the 3 components of their model. Why use 3 different things? Could you not have used B-splines for all 3? It seems to make the model overly-complex this way.
- You used 20-30 ms time bins for the transformer. Why?
- Could the authors comment on how data hungry their approach might be compared to end alternative, non-transformer method?

**Ethical Concerns:**

["NO or VERY MINOR ethics concerns only"]

**Limitations:**

Yes

**Quality:**

3

**Strengths And Weaknesses:**

The paper was exceptionally easy to read, well organized, and very polished. The approach is novel and demonstrates excellent performance both in terms of ground-truth recovery  and interpretability. I think that the method could be very valuable to analysis of high-density recordings. I have no major criticisms but I do have some questions which I detail below.

---

> ### Author Rebuttal · Authors · 2025-07-31
>
> We thank the Reviewer pXQU for the helpful comments and for recognizing the value of our work. We address your questions below, and ***we will update the revision to provide more clarification***.
>
> > Q1) Why use a combination of splines, raised cosine basis functions, and “penalized roughness”?
>
> We agree this could benefit from additional explanation. In principle, several types of basis functions could be used for all components of our model, as they are often roughly equivalent in representational power. However, we chose different basis functions for specific reasons grounded in modeling convention and performance.
>
> We use raised cosine basis functions for coupling because they are denser at short time lags, which allows for rapid changes immediately following a spike. This aligns well with known properties of post-spike and coupling effects and follows established practices in GLM modeling [1].
>
> For individual-neuron dynamics, we use uniform B-spline basis functions, which do not have the inductive biases built into the raised cosine basis.
>
> Finally, we apply penalized roughness to encourage smoothness in the estimated individual-neuron dynamics. While it is possible to achieve smoothness using a small number of B-spline basis functions, we believe the penalty approach offers more systematic, automatic, and controllable regularization, as discussed in the spline-based modeling literature [2,3].
>
> > Q2) Why use 20-30 ms time bins for the transformer
>
> Thank you for bringing this up! We tried shorter or longer time bin widths; the performance (log likelihood) drops slightly if using time bins bigger than 20ms or smaller than 10ms. The model is relatively insensitive within the 10–20 ms range, and we found no substantive differences between these settings. A possible explanation is that larger time bins lose temporal information, while very small bins require more data to train and may not carry too much additional information.
>
> Below is a table showing the relative negative log-likelihood (NLL) of the model across different time bin widths for each dataset. Values are offset by the best (minimum) NLL for each dataset, so lower is better, and 0 indicates the best performance.
>
> Although the 20 ms bin width appears to perform better than 10 ms in this table, the results here were obtained on a smaller dataset ( 2000, 2000, and 5915 trials for the three datasets, respectively) due to limited time during the rebuttal phase. In contrast, our formal hyperparameter tuning (performed on a larger real dataset)  selected 10 ms as the optimal bin width, which is what we used in the final model.
>
> | Time bin width (ms)  | NLL on GLM synthetic dataset  | NLL on EIF synthetic dataset | NLL on Allen Institute dataset
> | ------------- |:-------------:| ------------- |:-------------:|
> | 2      | 2.43e+04     | 3.39e+04      | 6.39e+04     |
> | 5      | 1.09e+04     | 3.07e+04      | 6.30e+04     |
> | 10      | 1.43e+02     | 9.56e+01      | 1.42e+04     |
> | 20      | 0     | 0      | 0     |
> | 40      | 2.21e+03     | 6.03e+04      | 1.35e+04     |
>
> > Q3) how data hungry compared to non-transformer method
>
> This is a great point and we will add a comment to the results in the revised manuscript. We evaluated model performance as a function of training set size using our EIF synthetic dataset, comparing the GLM-Transformer with a simpler non-Transformer GLM variant. Both models eventually reach a performance plateau, but the simpler GLM converges quickly with as few as 100 training trials. In contrast, the GLM-Transformer generally requires around 1500 trials to reach a higher-performing plateau.
>
> This higher data demand reflects the flexibility of the Transformer-based component and its capacity to model complex trial-to-trial variability. One possible future direction when encountering smaller dataset is to pretrain a foundation model across multiple neural datasets, then fine-tuning on the smaller one.
>
> ### References
>
> [1] Pillow J W, Shlens J, Paninski L, et al. Spatio-temporal correlations and visual signalling in a complete neuronal population[J]. Nature, 2008, 454(7207): 995-999.
>
> [2] Hastie T, Tibshirani R, Friedman J H, et al. The elements of statistical learning: data mining, inference, and prediction[M]. New York: springer, 2009.
>
> [3] Hastie, T., Tibshirani, R., & Friedman, J. (2009). *The Elements of Statistical Learning: Data Mining, Inference, and Prediction* (2nd ed.). Springer.

---

> > ### Comment · Reviewer_pXQU · 2025-08-04
> >
> > thank you! My concerns are satisfied.

---

> > > ### Author Response · Authors · 2025-08-05
> > >
> > > We are truly grateful for the positive assessment you have given to our work! Your comments have already helped us identify ways to improve the paper, and we look forward to incorporating them in the revision.
> > >
> > > Thank you again for your suggestions and comments.

---

### Note · Authors · 2025-08-14

We thank the reviewers and AC for their constructive comments and suggestions throughout the review process. We sincerely appreciate the feedback; it will help us clarify our positive contribution along with the limitations of our implementation.

On the neuroscience side, our work emphasizes the rich, trial-varying individual-neuron dynamics that often confound major effects of interest, especially cross-population coupling. Our model explicitly addresses this issue by separating background dynamics from population-level coupling. We believe this disentanglement is critical for obtaining more accurate and interpretable measures of cross-area interactions.

On the methodology side, we see combining the flexibility of deep learning with the interpretability of GLMs as a promising direction for neural data analysis. GLM-Transformer demonstrates this idea by using a deep latent model to capture individual-neuron dynamics, while maintaining interpretability through a structured coupling component. This allows us to directly estimate the effect of spikes from one population on the firing rate of another population.

While the deep latent model component builds on existing architectures, the novelty of our work lies in formulating a challenging and underexplored problem, and in developing a principled solution that combines the complementary strengths of deep learning and GLMs in a coherent and interpretable framework.

We will revise the manuscript to better highlight the scope of our contribution, address remaining concerns (e.g., the  fixed-length-trial assumption, the empirical role of the Transformer encoder, and the framing of novelty), and incorporate additional analyses requested during the rebuttal. We sincerely thank the reviewers and AC again for their time, thoughtful feedback, and guidance.

---

### Decision · Program_Chairs · 2025-09-17

**Decision:**

Accept (poster)

**Comment:**

This paper proposes GLM-Transformer, a hybrid framework that combines a Transformer-based VAE to capture trial-to-trial, neuron-specific variability with a GLM-style coupling component to identify interpretable cross-area interactions. The approach directly addresses a key confound in multi-area neural data analysis: background fluctuations that can obscure true directed connectivity. Through simulations, mechanistic models, and application to the Allen Institute Visual Coding dataset, the method is shown to recover known hierarchical pathways while remaining robust to shared variability.

Reviewers consistently found the paper well-written, technically sound, and relevant, highlighting the creative integration of GLMs and deep latent models. Strengths noted include the principled separation of nuisance dynamics from coupling, the biological interpretability of the filters, and the robustness demonstrated across datasets. The authors’ rebuttal clarified several concerns, adding analyses on larger populations, new simulations with four areas, and expanded discussion of assumptions and related work. These additions resolved most critical questions, and two reviewers raised their scores after rebuttal.

The main reservations concern the degree of methodological novelty, the limited advantage of the Transformer encoder over simpler RNNs in ablations, and the fixed-trial-length assumption, which could restrict applicability. Still, the consensus is that the paper makes a meaningful contribution by framing and solving an important challenge with a practical, interpretable, and scalable approach. On balance, I recommend acceptance.